# Non-dominant hand contractions do not facilitate performance under pressure in common desktop tasks

Yu Fan Eng[1], Daniel R. Little[1], Andy Yang[2], Anchalee Wensinger[2], Leo J. Roberts[2,3]*

**1** School of Psychological Sciences, University of Melbourne, Australia, **2** School of Psychological Sciences, Monash University, Australia, **3** Melbourne School of Population and Global Health, University of Melbourne, Australia

* leo.roberts@unimelb.edu.au

**Data Availability Statement:** The data files that support this publication are available in an Open Science Framework public repository (DOI: 10. 17605/OSF.IO/GY2F9).

## Abstract

Non-dominant hand contractions (NDHCs) have been shown to help expert motor skills in high-pressure scenarios that induce performance anxiety. Most studies of NHDCs under pressure have examined benefits in overlearned specialist movements (e.g., sporting skills), while few have considered if NDHCs can aid common movements with population-wide expertise (e.g., typing). Accordingly, across three experiments, we explored if NDHCs could protect or facilitate performance under time and/or evaluation pressure in a cursor positioning task (Experiments 1 & 2) and a typing task (Experiment 3). Despite varying the nature of the task, pressure manipulation, and design, and successfully manipulating state anxiety in each experiment, we found no evidence that NDHCs assist performance under pressure in these tasks. For the pressure × contraction condition interaction, the largest inclusion Bayes Factor was .40 for task response time and .62 for task error (Experiment 1), indicating evidence in favour of a null result. Our results, along with other recent studies in this area, cast doubt on the benefits of NDHCs under pressure outside sporting tasks and underline the need for a better mechanistic account of the phenomenon.

## Introduction

Performance anxiety is a common psychological issue driven by deeply rooted concerns about appearing incompetent [1]. When a performance has meaningful consequences and/or time limits are imposed (i.e., there is pressure), anxiety can grow and the task can become more challenging due to attentional disturbances, such as interfering self-conscious thoughts [2, 3]. Without self-regulation, performance can decline from baseline [4], and in the extreme, dramatic underperformance can occur, known as choking under pressure [5]. To achieve their goals, performers need practical ways to manage anxiety when preparing for or participating in important events.

Research on the impact and treatment of performance anxiety in motor skill domains has mostly focused on specialist skills (e.g., sporting movements). Nevertheless, ordinary, well-rehearsed motor tasks (e.g., typing) are also vulnerable to performance anxiety induced by

**Funding:** The author(s) received no specific funding for this work.

**Competing interests:** The authors have declared that no competing interests exist.

time limits, higher stakes, or audience observation. For instance, elevated pressure to perform can interrupt index finger movements [6]. Relatedly, anxiety about falling can harm walking stability [7]. It is therefore worthwhile to consider the effectiveness of performance anxiety treatments in ordinary motor tasks as well as in specialist pursuits. Doing so would access a huge pool of 'expert' participants and facilitate an expanded evaluation of treatments.

Non-dominant hand contractions (NDHCs)–such as the contraction of the left hand by a right-handed individual–have been presented as a promising treatment for performance anxiety for athletes facing high-pressure situations [8]. To date, simply squeezing a soft ball with the non-dominant hand for 30–45 seconds just prior to a high-pressure event has revealed performance benefits in five sports [9–11]. NDHCs may facilitate performance by suppressing conscious interference before or during movement [12, 13], allowing a more automatic execution. Despite the potential of NDHCs to assist any number of pressure-affected motor skills, few have studied their value outside sport or music.

## Hemisphere-specific priming

Hand contractions are believed to assist motor skill performance under pressure by a process of hemisphere-specific activation, where one of the brain's hemispheres becomes relatively active following contralateral hand movement [14]. Using Electroencephalography (EEG) recordings, several investigations have revealed that contracting one hand leads to a relatively large reduction in alpha band power (representing elevated cortical activity) over the primary motor (and other) areas of the contralateral hemisphere [15–17].

Various studies have also found that selective hemispheric activation, via hand contractions, is followed by emotional and behavioural changes consistent with proposed functions of the preferentially activated hemisphere [14, 16, 18–22]. For example, Goldstein, Revivo (21) observed that left hand contractions (leading to right hemisphere activation) enhanced participants' performance in the Remote Associates Test [23]. This test is a convergent and divergent thinking task, thought to index creative thinking, in which participants try to generate one word that is related to three presented words (e.g., the word 'sweet' might be used as a solution for the words 'tooth, potato and heart'). The right hemisphere has been implicated in doing an adapted version of this task [24] and is generally regarded as more dominant in creative thinking activities [25].

It remains unclear how nebulously biasing activity towards one hemisphere can result in specific behavioural changes. There is also doubt over the principle of biased contralateral activation following hand contractions. Cross-Villasana, Gröpel [13] found a comparable increase in cortical activity over motor areas in *both hemispheres* following NDHCs, and then a prolonged reduction in cortical activity across the whole scalp once contractions ceased–a finding replicated by Mirifar, Cross-Villasana [26].

## Benefits of NDHCs in skilled motor performance under pressure

Seminally, Beckmann et al. (2013) proposed that NDHCs could help motor skill execution under pressure, grounded in the idea that boosting right-hemisphere activity would encourage an ideal brain state for expert motor execution. This expert state (at least in aiming tasks) is characterised by (a) suppressed activity in left hemisphere temporal brain sites associated with verbal-analytic thought and (b) increased activity in right hemisphere brain sites associated with visual-motor processes [27–29]. The notion that motor skill experts have less verbal-analytic engagement is consistent with well-known skill acquisition theories [30, 31]. These accounts share that learners rely on effortful, verbal, and conscious skill execution as novices,

before progressing to relatively effortless, non-verbal, and autonomous skill execution as experts.

Importantly, many studies indicate that when experts engage in verbal-analytic processes (e.g., thoughts related to how to execute a movement, technical rules, monitoring the movement closely), performance is likely to decline [see 32 for a review]. This finding is consistent with self-focus theories of choking, which share that pressure can encourage a performer to pay conscious attention to the execution process, resulting in novice-like, verbal-analytic processing that is too slow to produce expert-level movement [3, 33, 34]. Beckmann et al (2013) reasoned that priming the right hemisphere before a high-pressure sporting moment might help re-establish an expert brain state, destabilised by pressure, by boosting visuo-motor processing in the right hemisphere and thus preventing the verbally oriented left hemisphere from dominating execution.

To better understand the impact of hand contractions on brain activation patterns in motor preparation, Hoskens, Bellomo [12] took EEG readings while novice participants prepared for golf putts following periods of NDHCs, DHCs, and sitting quietly. Compared to DHCs and sitting quietly, NDHCs resulted in less connectivity (or co-activation) between a left hemisphere temporal site (T7) thought to index verbal-analytic activity and a frontal midline site (Fz) thought to index motor planning. Several other investigators have zeroed in on the possibility that this T7-Fz coherence indexes the extent of verbal-analytic engagement in movement, where lower levels of engagement reflect greater expertise [35] and predict better motor executions [36]. Hosken et al.'s results are consistent with the idea that NDHCs reduce verbal-analytic interference during motor planning.

## NDHCs in sport

Neurological mechanisms aside, several studies have revealed beneficial effects of NDHCs on performance under pressure in sport [9–11]. Originally, Beckmann, Gröpel and Ehrlenspiel [9] separately investigated the effects of hand contractions on soccer penalty shots, taekwondo kicks, or badminton serves performed under manipulated pressure. Just prior to a high-pressure task, right-handed athletes were randomly assigned to squeeze a soft ball for 30 seconds in their dominant or non-dominant hand. Each time, the investigators found that the athletes who contracted their dominant hand showed declined performance under pressure, while those who contracted their non-dominant hand did not. Beckmann, Fimpel and Wergin [11] replicated the effect; junior tennis players who made pre-performance NDHCs served better under competition pressure than those who contracted the dominant hand. Quasi-experimental work by Gröpel and Beckmann [10] reached the same conclusion. Artistic gymnasts who made NDHCs before a routine performed better than gymnasts who used typical preparations. As a point of difference, a recent randomised control trial with aspiring musicians found no performance benefits following NDHCs in a high pressure mock performance [37].

To date, researchers have considered the after-effects of NDHCs benefits in elite sport and music, with little consideration of the after-effects in non-elite domains. Accordingly, we tested the benefits of NDHCs on well-known ("everyday") computer tasks conducted under time and/or performance pressure. There are two advantages of using everyday computer tasks. First, numerous skilful individuals are available, who reliably practice computer-based motor actions in their job, education, or social communications. Second, if NDHCs can facilitate performance in ordinary computer tasks under pressure, the utility of NDHCs could be broadened to domains like computer-based testing in academic or job-interview settings, office work under deadlines or competitive computer game playing.

## Present study

We examined the protective effects of NDHCs in three experiments, first in a cursor positioning task (Experiment's 1 and 2) and then in a typing task (Experiment 3). In Experiment 1, an NDHC group and a 'sitting quietly' group performed a speed accuracy cursor positioning test using the Bullseye Task [38] under conditions of low time-pressure and high time-pressure. In Experiment 2, we again used the Bullseye Task but manipulated pressure more comprehensively (with competitive, monitoring and time pressure), and added an intermediary high-pressure condition before introducing the NDHC (or sit quietly) manipulation. In Experiment 3, we tested the effect of hand contractions in another ordinary motor skill–typing. Here, an NDHC group, a DHC group and a sit-quietly group typed passages of text as quickly and accurately possible under low and high pressure (mixed competitive, monitoring and time pressure).

## Experiment 1

In Experiment 1, we investigated the after-effects of NDHCs in a cursor positioning accuracy task (the Bullseye Task). Right-handed participants first completed the task in a low-pressure situation (no time limit), then made NDHCs or sat quietly for 30s according to random assignment, then completed the task in a high-pressure situation (stringent time limit). While much NDHC research has included an active control condition (i.e., a group making DHCs), we did not. This was due to a concern that DHCs would fatigue the cursor positioning hand (the right hand) and encourage a positive NDHC effect. The absence of a DHC group did not concern us greatly. In the first instance, we sought evidence that NDHCs at least offered some benefits compared to sitting quietly.

Since NDHCs are believed to buffer the de-automotive effects of pressure-induced self-focus (i.e., verbal-analytic intrusion) and/or encourage automatic execution, it follows that the protective benefits of NDHCs under pressure are most easily observed in automatic motor tasks. We concluded that cursor-positioning would be a relatively automatic movement for our sample of computer-literate university-aged people because it is simple ballistic task that requires minimal explicit instruction or thought and would have been heavily rehearsed. Based on the protective effects of NDHCs observed in sporting movements under pressure, we expected that the NDHC group would be better protected under pressure than the sitting quietly group.

The first experiment was pre-registered on osf.io (https://osf.io/jnv4q). To maintain conciseness, minor modifications were made in the reporting of results.

## Method (Exp 1)

### Participants

A large effect of NDHCs on performance was estimated [9, 10]. Power analysis using G*Power indicated that a sample size of 16 could detect a significant interaction ($p < 0.05$) in a $2 \times 2$ between-within design [39]. Ultimately, 111 participants (77 Female, 31 Male, three who did not identify as female or male) aged 17–38 years ($M = 19.29$, $SD = 2.43$) were recruited through an undergraduate psychology research experience program (REP). We excluded 19 left-handed participants and two people with incomplete data (technical difficulties). Three additional participants were lost due to insufficient data following removal of (a) single-trial error outliers (three SDs > mean), (b) participant-level error outliers, and (c) single trial RTs below 200ms and above 3000ms (deemed unreasonably fast/slow). The final sample had 87 right-handed participants (60 Female, 25 Male, two whom did not identify as female or male) aged 17–26 years ($M = 19.10$, $SD = 1.54$).

Several device types were used to complete the experiment. Sixty-one participants used a trackpad, 23 used a classic mouse, two used a touchscreen device, and one used a rollerball mouse. Device usage was unbiased across the NDHC group ($N$ = 41; 30 Trackpad, 11 Classic Mouse) and the sitting quietly group ($N$ = 46; 31 Trackpad, 12 Classic Mouse, one Rollerball Mouse, two Touchscreen Device). Rollerball mouse and touchscreen device users were retained since their exclusion did not impact the results. The experiment was approved by The University of Melbourne Human Research Ethics Committee (HREC Approval Number 2056768.1) under guidelines that allow recruitment of minors with sufficient maturity and understanding to consent (hence separate parental consent was not sought nor obtained for individuals under 18). Data was collected from May 18 to June 15, 2020.

## Stimuli and apparatus

The stimulus was a bullseye target on a black background with nine white concentric circles [see Fig 1, 38]. The innermost circle denoted the centre of the target and was red (RGB = [255, 109, 120]). The bullseye was either presented in the upper left or upper right corner of the screen, with the target centre at 75% screen height and 25% screen width.

The experiment was conducted online during the COVID-19 pandemic (in person experimentation was not possible). The experiment was browser-based and written in Javascript using jsPsych [40] and HTML. Participants were provided a link to complete the experiment on their device, unsupervised. Participants were asked to remove distractions at experiment outset and indicate the device used (e.g., standard mouse). Pictures of possible devices were presented to assist selection. Participants were asked to obtain a rolled-up pair of socks for the NDHC manipulation. Previous work employed stress balls; however, our online study required a different stimulus since not all participants could have accessed a stress ball.

## Instruments

**Edinburgh Handedness Inventory-Short Version (EHI-SV).** The EHI-SV [41] assessed handedness at experiment outset. The EHI-SV has four items on a five-point Likert scale. Participants indicated hand usage preference in activities with "Always Left", "Usually Left", "Both Equally", "Usually Right", or "Always Right" (respective score of -100, -50, 0, 50, and

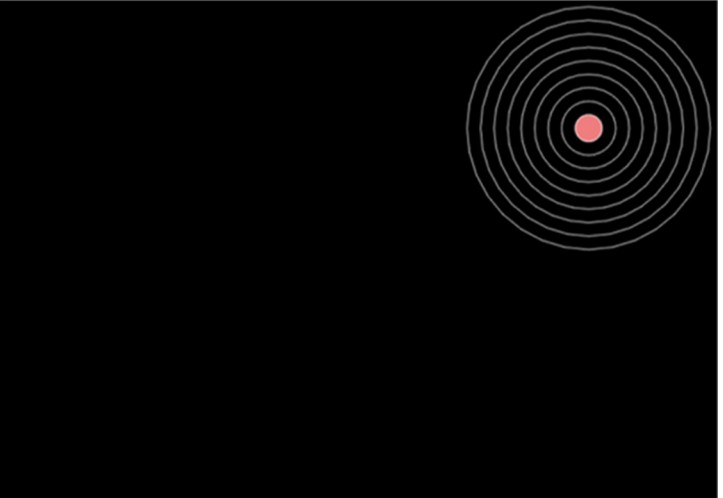

**Fig 1. Bullseye target stimulus.**

100). An average score $< = 61$ indicates right-handedness (those scoring $< 61$ were removed). The inventory has good internal consistency ($\alpha = .93$) [41].

**Mental Readiness Form-Likert (MRF-L).** The MRF-L [42] measured state anxiety. It was selected for robustness against social desirability bias, shortness, and systematic scoring method. The MRF-L has three items, each representing cognitive anxiety (CA), somatic anxiety (SA), and self-confidence (S-C). Each item is presented with a 11-point bipolar Likert scale, with "calm" to "worried" for CA, "relaxed" to "tense" for SA, and "confident" to "scared" for S-C. Items are scored and analysed individually. The MRF-L has adequate internal consistency ($\alpha = .74 - .87$) [43].

## Design and procedure

The experiment had one session (~30-minutes). To begin, participants reported the input device used, calibrated their screen, and completed the EHI-SV. During calibration, participants clicked the centre of the bullseye (a small black dot inside the red bullseye) ten times on each side (i.e., upper left and upper right-hand corners of the screen). The mean location of clicked points was used as the true location of the bullseye for each participant, ensuring bullseye location accuracy across different monitor resolutions and display sizes. The black dot was removed after calibration.

The experiment consisted of an Accuracy block (no time pressure) and a Speed block (time pressure), both containing six practice and 30 experimental trials. The Accuracy block occurred before the Speed block. In the Speed block, participants were randomly assigned to sit quietly or squeeze a ball of socks with their left-hand (NDHCs) for 30 seconds between the practice and experimental trials, using an on-screen countdown clock. Prior to the manipulation, participants were (deceptively) told that their assigned activity is known to improve performance in high pressure tasks.

Before each block, participants were reminded of their condition (Speed/Accuracy). Participants then clicked a fixation cross to start the trial, centring the cursor on the screen. A fixed time interval of 500ms occurred before the bullseye appeared on screen. The location of the bullseye (left or right) was determined randomly on each trial with an equal number of left and right locations in both Accuracy and Speed blocks. Participants were given either 750ms (Speed block) or unlimited time (Accuracy block) to click on the middle of the bullseye (i.e., the red circle). Participants were required to make a response even when the bullseye stimulus was removed after 750ms. Response time (RT) and error (i.e., distance from target) were recorded regardless of meeting the deadline or not. If a participant did not respond in time (speed condition only), feedback stating 'Too Slow! Respond Faster!' appeared for 1000ms after the response. A fixed time interval of 500ms occurred again before the next trial. Fig 2 displays the trial procedure. The MRF-L was completed four times: before and after each experimental block as in Beckmann, Gröpel and Ehrlenspiel [9].

## Data analysis

Error was calculated as the Euclidean distance between the clicked point and the true centre of the bullseye for each trial. Analysis was conducted on normalised Euclidean distance by correcting the error based on the size of the browser window (recorded using JavaScript commands). RTs were recorded from the moment that participants clicked the fixation cross until the moment they made a response. To examine the effects of NDHCs on performance, we conducted a Bayesian Between-Within 2 Contraction Condition (NDHCs vs Sit Quietly) × 2 Pressure Condition (Accuracy vs Speed) Analysis of Variance (ANOVA) for both mean RTs and Error using JASP [44].

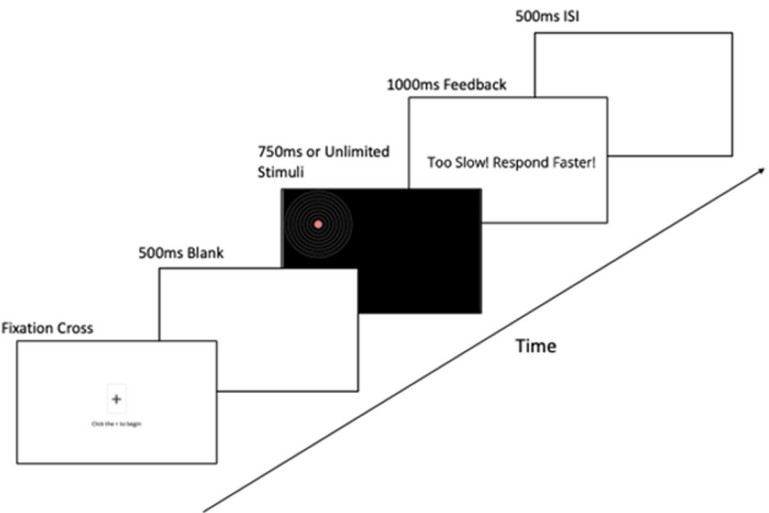

**Fig 2. Display of trial procedure.**

State anxiety was analysed (as a proxy for pressure) with a Bayesian One-Sided Paired Samples t-tests performed in JASP. We compared all MRF-L domain scores (CA, SA, S-C) recorded between completion of the Accuracy block (when participants best understood the low-pressure task) and completion of the Speed practice trials (when participants were first aware of the demands of the high-pressure task). We expected relative increases across all three domains in the speed compared to accuracy condition.

All inferences were made using Bayes Factors (BF). BF are measures of relative predictive performance of the alternate hypothesis ($H_1$) over the null hypothesis ($H_0$). Thus, the Bayes factors (for comparing two competing models in t-tests and post-hoc tests; $BF_{10}$) and inclusion Bayes factors for main and interaction effects (for assessing the inclusion of predictors in ANOVA; $BF_{incl}$) quantify the degree to which the data supports either hypothesis. Specifically, BF values $> 1$ indicates increased support for $H_1$ while BF $< 1$ indicates increased support for $H_0$. We used the interpretation scheme for BF values used by Lee and Wagenmakers [45] (see Table 1).

## Results (Exp 1)

### Pressure manipulation

Bayesian One-Sided Paired Samples t-tests revealed extreme evidence for an increase in CA, an increase in SA, and a decrease in S-C, indicating that the time pressure manipulation was successful (Table 2).

### RT

Mean RTs by Pressure and Contraction condition are presented in Table 3. The 2×2 Bayesian ANOVA on RTs (Table 4) revealed extremely strong evidence for a main effect of Pressure ($BF_{incl} > 100$), reflecting faster trial completion in the Speed condition than the Accuracy condition. There was moderate evidence against a main effect of Contraction condition and anecdotal evidence against a Pressure × Contraction interaction. In summary, the decrease in RTs under time pressure did not differ between the NDHC and sit-quietly groups.

**Table 1. Classification scheme of $BF_{10}$ [45].**

| Bayes Factor ($BF_{10}$) | Evidence Category |
|---|---|
| > 100 | Extreme evidence for $H_1$ |
| 30–100 | Very strong evidence for $H_1$ |
| 10–30 | Strong evidence for $H_1$ |
| 3–10 | Moderate evidence for $H_1$ |
| 1–3 | Anecdotal evidence for $H_1$ |
| 1 | No evidence |
| 1/3–1 | Anecdotal evidence for $H_0$ |
| 1/10–1/3 | Moderate evidence for $H_0$ |
| 1/30–1/10 | Strong evidence for $H_0$ |
| 1/100–1/30 | Very strong evidence for $H_0$ |
| < 1/100 | Extreme evidence for $H_0$ |

### Errors

The 2×2 ANOVA on error scores (Table 4) revealed extremely strong evidence for a main effect of Pressure, reflecting greater error in the Speed condition than the Accuracy condition (Table 3). In contrast, there was anecdotal evidence against a main effect of Contraction condition and a Pressure × Contraction interaction. In summary, the increase in error scores under time pressure did not differ between the NDHC and sit-quietly groups.

## Discussion (Exp 1)

Experiment 1 explored the protective effects of NDHCs in a cursor positioning task performed under time-pressure. The results did not support the hypothesis that NDHCs would protect performance under time pressure. Participants in the NDHC condition performed no differently to those who sat quietly; both groups demonstrated a comparable increase in error scores and decrease in RTs under time pressure. These results are inconsistent with the findings of similar studies conducted with athletes.

We successfully induced time pressure in this experiment (increased MRF-L scores). By contrast, Beckmann et al. (2013) induced pressure with competition and potential rewards. This methodological difference might have influenced the protective effects of NDHCs. Additionally, participation in the Experiment 1 was unmonitored. Improper task completion (especially NDHCs) could have led to a reduced effect and hindered our ability to observe protected performance. Experiment 2 addressed both concerns.

## Experiment 2

Two adjustments in Experiment 2 allowed us to better test the after-effects of NDHCs in the Bullseye Task. First, we added a second Speed block that was carried out under competitive pressure (on top of the original Accuracy and Speed blocks from Experiment 1). In the second

**Table 2. One-sided paired-samples t-test on the MRF-L responses, M (SD).** Note that a higher score on the self-confidence domain denotes less self-confidence.

| | Accuracy | Speed | $BF_{10}$ |
|---|---|---|---|
| Cognitive | 2.67 (2.26) | 5.54 (2.63) | > 100 |
| Somatic | 3.07 (2.23) | 5.56 (2.37) | > 100 |
| Confidence | 2.33 (2.07) | 5.15 (2.49) | > 100 |

**Table 3. Performance descriptive statistics, M(SD).**

| Measure | Condition | Accuracy | Speed |
|---|---|---|---|
| RT | LH Contraction | 1087.53 (295.16) | 695.01 (91.62) |
| | Sit Quietly | 1116.10 (418.26) | 648.11 (85.37) |
| Error | LH Contraction | 5.63 (3.19) | 22.64 (11.23) |
| | Sit Quietly | 6.10 (3.25) | 27.54 (16.06) |

Speed block, we offered monetary prizes for the best-performing participants (relative to the first Speed block) and displayed a pre-competition leaderboard that, regardless of performance, indicated an opportunity to win a prize. Second, the entire experiment was conducted over a shared Zoom screen. This allowed us to confirm the execution of NDHCs and passively apply audience/monitoring pressure. Audience monitoring should theoretically encourage conscious or self-focused processing under pressure, which is precisely the kind of attentional interference that NDHCs are believed to counteract [9].

The addition of the second Speed block also gave us a baseline of Speed performance to compare against (see Beckmann, Gröpel and Ehrlenspiel [9] Experiment 3 for a similar approach). In Experiment 1, we effectively forced inferior Bullseye Task performance via stringent time limits. By adding this second Speed condition with competition/monitoring pressure, we could examine if participants underachieved against their baseline Speed performance, on account of competition/monitoring pressure. Under this scenario, we could more directly examine the ability of NHDCs to protect against choking. We once again predicted that NDHCs made before the Bullseye Task was completed under time, competitive, and monitoring pressure would protect performance (compared to sitting quietly).

## Method (Exp 2)

### Participants

A hundred and fourteen participants (66 Females, 44 Males, four people who did not identify as female or male) between 16 and 51 years of age ($M = 24.52$, $SD = 6.51$) were recruited. We excluded participants who were left-handed ($N = 24$), had incomplete data due to technical difficulties (N = 4), or both ($N = 2$). Four additional participants were lost due to insufficient data following the removal of outliers as per Experiment 1. We removed a final participant who did not read the experiment instructions.

The final sample comprised 79 students (50 Female, 27 Male, 2 who did not identify as female or male) aged 16–54 years ($M = 24.87$, $SD = 7.53$). Device usage frequencies were evenly distributed between the NDHC group ($N = 38$; 20 Mouse, 18 Trackpad) and the sitting quietly group ($N = 41$; 22 Mouse, 18 Trackpad, one reported "Other").

**Table 4. Bayesian ANOVA results for RT and error.**

| Measure | Effects | P(incl\|data) | BF$_{incl}$ |
|---|---|---|---|
| RT | Pressure | .94 | > 100 |
| | Contraction | .16 | .21 |
| | Pressure × Contraction | .06 | .40 |
| Error | Pressure | .81 | > 100 |
| | Contraction | .30 | 0.60 |
| | Pressure × Contraction | .19 | 0.62 |

All participants recruited from the REP system were reimbursed one course credit. The best performing participant across our two high pressure conditions was given a $100 gift card. Testing was approved by The University of Melbourne Human Research Ethics Committee (HREC Approval Number 2056768.2) under guidelines that allow recruitment of minors with sufficient maturity and understanding to consent (hence separate parental consent was not sought nor obtained for individuals under 18). Data was collected from August 16 to September 24, 2020.

## Design and procedure

Experiment 2 comprised one session (~35 minutes) held over Zoom. Participants were on camera throughout and shared their screen (the experiment webpage). At experiment outset, participants reported the type of device used, calibrated their screen, and completed the EHI-SV.

The session had three blocks (one Accuracy, two Speed), with the Accuracy block occurring first. The Accuracy block and Speed block 1 had six practice and 30 experimental trials, while Speed block 2 only had 30 experimental trials. Participants completed the MRF-L at five time-points: after Accuracy practice trials (Time Point 1; TP1), after Accuracy experimental trials (TP2), after Speed block 1 practice trials (TP3), after Speed block 1 trials (before leaderboard presentation; TP4), and after Speed block 2 trials (TP5).

Prior to Speed block 1, participants were informed about a monetary prize for the best-performing participant. Participants were also told that a leaderboard would display their error scores and ranking after the completion of Speed block 1. No NDHCs were performed at this time. After completing the MRF-L, each participant's scores were displayed on a fake leaderboard, which ranked them at third place regardless of performance. The goal was to make participants think that they were close to the top but needed to perform well in Speed block 2 to win the monetary prize. Participants' error scores on the leaderboard were adjusted to reflect a plausible third place ranking if their actual scores were too close to the fake top or bottom scores. Prior to Speed block 2, participants were randomly assigned to sit quietly or make NHDCs with a pair of socks for 30 seconds. Participants were told that their assigned activity is known to improve task performance in high pressure tasks.

## Data analysis

Error and RT were calculated as in Experiment 1. To examine the effects of NDHCs on performance, we conducted a Bayesian Between-Within 2 Contraction Condition (NDHCs vs Sit Quietly) × 3 Pressure Condition (Accuracy vs. Speed 1 vs. Speed 2) ANOVA for both mean RT and error in JASP. We expected a main effect for Pressure in both ANOVAs, and an interaction effect for the Error ANOVA.

State anxiety (pressure induction) was analysed with a Bayesian Repeated Measures ANOVA performed in JASP. Given that there was a main effect of Pressure, post-hoc comparisons were performed across the three blocks for each MRF-L domain. We expected to see differences between (a) TP2 and TP3, (b) TP2 and TP4, and (c) TP3 and TP4.

## Results (Exp 2)

### Pressure manipulation

Descriptive statistics implied increased state anxiety under time pressure (Table 5). The Bayesian repeated measures ANOVA on state anxiety confirmed this, revealing extreme evidence for a main effect of Pressure in each domain (Table 6). Post-hoc comparisons revealed extreme

**Table 5.  MRF-L scores TP2:TP4, M(SD).**

|  | TP2 | TP3 | TP4 |
|---|---|---|---|
| Cognitive | 2.65 (2.29) | 5.20 (2.56) | 5.22 (2.54) |
| Somatic | 3.09 (2.31) | 5.20 (2.55) | 5.52 (2.54) |
| Self-Confidence | 2.52 (2.03) | 4.89 (2.21) | 4.90 (2.31) |

differences between TP2 and TP3, and TP2 and TP4 timepoints, suggesting that both pressure manipulations increased state anxiety compared to the accuracy condition (Table 7). There was anecdotal to moderate evidence against a difference between the TP3 and TP4 timepoints, indicating that state anxiety was sustained after the first speed block. State anxiety patterns did not differ between contraction conditions, reflected by anecdotal to moderate evidence against a Contraction effect (Table 6).

### RT

Mean RTs for each Pressure and Contraction condition are in Table 8. The 2×3 Bayesian ANOVA on RT revealed extremely strong evidence for a main effect of Pressure (see Table 9). Post-hoc comparisons revealed extreme evidence for a decrease in RTs between the accuracy block and both speed blocks (see Table 10). These comparisons also revealed very strong evidence for reduced RTs between Speed block 1 and Speed block 2. We found moderate evidence against a Contraction main effect and very strong evidence against the Pressure × Contraction interaction, implying that RT decreases did not differ between the NDHC and sit-quietly groups (Table 9).

### Error

The 2×3 Bayesian ANOVA on error revealed extremely strong evidence for a main effect of Pressure (Table 9). Post hoc analyses (Table 10) revealed extremely strong evidence for differences between the accuracy block and both speed blocks, reflecting higher error under time pressure (Table 8). There was moderate evidence against error differences between Speed block 1 and Speed block 2. We also found moderate evidence against a main effect of Contraction condition and strong evidence against a Pressure × Contraction (Table 9). This indicates that error patterns between Pressure conditions did not differ between the NDHC and sitting quietly groups.

## Discussion (Exp 2)

In Experiment 2, we examined the protective effects of NDHCs on Bullseye Task performance under time, monitoring, and competitive pressure. Here, participants undertook one accuracy and two speed blocks, with the NDHC manipulation (NDHCs or sitting quietly) administered between the first and second speed block. Both groups responded with faster RTs and greater error in the first speed condition than the accuracy condition. However, the NDHC

**Table 6.  Bayesian repeated measures ANOVA results for MRF-L scores.**

|  | Cognitive | | Somatic | | Self Confidence | |
|---|---|---|---|---|---|---|
|  | P(incl\|data) | BF$_{incl}$ | P(incl\|data) | BF$_{incl}$ | P(incl\|data) | BF$_{incl}$ |
| Pressure | .99 | >100 | .99 | >100 | .98 | >100 |
| Contraction | .25 | .33 | .22 | .29 | .27 | .38 |
| Pressure × Contraction | .01 | .03 | .01 | .06 | .02 | .06 |

**Table 7. Post-hoc comparisons in MRF-L scores.**

| State Anxiety Measurement | | Cognitive | | Somatic | | Self Confidence | |
|---|---|---|---|---|---|---|---|
| Time-point 1 | Time-point 2 | Posterior Odds | $BF_{10}$ | Posterior Odds | $BF_{10}$ | Posterior Odds | $BF_{10}$ |
| TP2 | TP3 | >100 | >100 | >100 | >100 | >100 | >100 |
| | TP4 | >100 | >100 | >100 | >100 | >100 | >100 |
| TP3 | TP4 | .07 | .12 | .50 | .85 | .07 | .12 |

manipulation did not differentially affect error or RT scores between the NDHC and sitting-quietly groups under higher (time or competitive) pressure, hence, these results do not indicate beneficial effects of NDHCs under pressure.

While Experiment 2 manipulated pressure in a similar way to previous studies of NHDCs in sport, other issues could have hampered observation of NDHC benefits. First, we lacked specific data to confirm that the cursor movement used in the Bullseye Task was sufficiently automated in our participants to observe the NDHC protection effect. Second, we could not use an active control to assess relatively impacts between NDHCs and DHCs. We conducted a final experiment that addressed both issues.

## Experiment 3

In Experiment 3, we examined the effect of NDHCs in a keyboard typing task. Typing is a necessary modern skill and accomplished performers are ubiquitous. It is estimated that most university-aged students have typing skills roughly equivalent to yesteryear professional typists [46]. The dual-handed nature of typing allowed us to examine the protective effects of NDHCs compared to DHCs, without concern that DHCs would bias the results. Furthermore, significant research indicates that skilled typing is automatic after word selection, suggesting that it is a suitable task to examine protective benefits of NDHCs under pressure.

A body of work by Logan, Crump and others advances the automatic nature of typing after word selection. Logan and Crump [47] proposed that the cognitive process of typing has a hierarchical structure, in which an outer loop produces a string of words to be typed while an inner loop produces–word by word–the keystrokes required. Several pieces of evidence support that inner-loop keystroke production is automatic. First, skilled typists exhibit little explicit knowledge of the location of keys [48], implying a procedural process without much conscious access [34]. In addition, skilled typists type words faster than non-words [49], implying that skilled typing involves recalling a learned collection of chunked sequences that puts little pressure on working memory and allows rapid processing; characteristics of other automatic skills [50]. Finally, when typists monitor their hands, performance suffers greatly [51], like other well-learned motor skills [52]. In theory, the inner loop can be directly accessed by providing typists with the words to type (obviating the involvement of the outer loop), resulting in automatic task for skilled typists.

**Table 8. Performance descriptive statistics, M(SD).**

| Measure | Condition | Accuracy | Speed 1 | Speed 2 |
|---|---|---|---|---|
| RT | NDHC | 1178.68 (328.97) | 686.37 (92.09) | 670.64 (86.16) |
| | Sit Quietly | 1183.63 (415.21) | 697.86 (103.38) | 676.67 (87.84) |
| Error | NDHC | 5.63 (3.10) | 27.52 (13.89) | 26.65 (14.35) |
| | Sit Quietly | 6.85 (4.97) | 28.44 (16.32) | 29.18 (17.60) |

**Table 9. Bayesian 2 x 3 ANOVA results for mean RT and Error.**

| Measure | Effects | P(incl\|data) | BF$_{incl}$ |
|---|---|---|---|
| RT | Pressure | .99 | > 100 |
| | Contraction | .16 | 0.19 |
| | Pressure × Contraction | .01 | 0.09 |
| Error | Pressure | .98 | > 100 |
| | Contraction | .22 | .29 |
| | Pressure × Contraction | .02 | .09 |

With a task optimised to observe the effects of NDHCs, we once again hypothesised that the NDHC group would experience performance buffering under pressure, this time when compared to DHC and sitting-quietly groups. Typing performance was quantified with a constructed words-per-minute (WPM) variable that captured typing speed and accuracy, as well as typing efficiency (correct keystrokes / total keystrokes). Our expectation was that the NDHC group exhibit better performance (WPM and efficiency) relative to a low-pressure baseline than the DHC or sit-quietly groups.

## Method (Exp 3)

### Design and participants

Forty-five participants (30 Male, 15 Female) aged 19–56 years of age (M = 25.53, SD = 7.74) took part in the study. Due to our relatively small sample size, we retained one left-handed and five ambidextrous participants. Four ambidextrous participants were assigned to the sitting quietly condition, while the other two contracted their right hand. A repeat analysis showed that their exclusion did not influence the interpretation of results. We have continued to use the NDHC, DHC labels for consistency. Recruitment relied on word-of-mouth strategies and digital flyers posted on social media. Ethical approval was granted from the Monash University Human Research Ethics Committee (Project ID: 27763). Data was collected from May 10 to June 20, 2021. Advertising stipulated that each participant needed to type frequently during the week, be an Australian resident (due to the nature of online prizes), be fluent in English, and be 18 years or older.

### Materials and task

Data was collected online on JsPsych and HTML-based platform, which delivered participants information and consent forms, a demographic and typing questionnaire and the typing experiment. The EHI-SV determined handedness, and the MRF-L captured state anxiety. The experimental task required participants to type five short paragraphs (~100 words) as quickly

**Table 10. Post-hoc comparisons between pressure conditions (RT, error).**

| Measure | Pressure Condition | | Posterior Odds | BF$_{10}$ |
|---|---|---|---|---|
| | Condition 1 | Condition 2 | | |
| RT | Accuracy | Speed 1 | > 100 | > 100 |
| | | Speed 2 | > 100 | > 100 |
| | Speed 1 | Speed 2 | 24.78 | 42.19 |
| Error | Accuracy | Speed 1 | > 100 | > 100 |
| | | Speed 2 | > 100 | > 100 |
| | Speed 1 | Speed 2 | .07 | .12 |

and accurately as possible. Paragraphs contained neutral content about border collies [53], stimuli used repeatedly by Logan, Crump and others [54]. We trimmed the original paragraphs to prevent fatigue and changed American spelling to suit Australian participants. The paragraphs were randomly selected without replacement from a pool of 10.

## Procedure

The study was conducted within a Zoom meeting arranged by the experimenter and participant. Once in the meeting, participants were sent a URL for the online study. Following consent, basic demographic information (age in years, gender and typing experience information were collected), and the EHI-SV was administered. Participants were given typing task instructions then two short practice typing trials, with the experimenter present on Zoom to answer questions.

**Low pressure block.** Next, participants turned off their camera and microphone inside the Zoom interface (experimenters did the same) before completing the MRF-L in private. Participants then typed four paragraphs (again, in private) that comprised the low-pressure block. Participants could take short breaks after each trial.

**High pressure block.** After typing the fourth paragraph, participants were entered into a competition that would be decided with one final paragraph. They were told that the three best performers (based on WPM) in this final trial (relative to their low-pressure average, i.e., the three biggest improvers) would win an Amazon Australia voucher for A$100, A$60, or $A40, respectively for first, second, and third place. Participants' low-pressure WPM average was presented on-screen at this time to remind them of their score to beat. Next, participants were told that they would be asked to turn on their Zoom video and audio and to share their Zoom screen throughout the competition trial. A cover story indicated that this was necessary so that experimenters could both observe how often the participant looked down at their fingers during typing and typing progress during the competition. The goals of our manipulation were to (a) raise the stakes of typing performance in a competition with prizes that any participant could win (no matter their previous performance), and in which there was a single opportunity; (b) bring participants out of privacy and expose their performance to an audience; and (c) heighten internal focus by noting that we would directly observer behaviour. Following the instruction of the high-pressure manipulation, the MRF-L was re-administered.

**Contraction conditions.** Upon sharing their screen and activating Zoom video and audio–but before commencing the competition–participants were randomly assigned to a NDHC, DHC, or sit-quietly group and received corresponding on-screen instructions. Guided by a 30 second countdown clock on the screen, the NDHC group contracted a ball of socks with their left hand; the DHC group contracted a ball of socks with their right hand; the sit-quietly group waited passively. Participants had socks ready at the experiment outset. All participants were told that their intervention was known to assist performance in competition. Post intervention, participants typed the competition paragraph, then were debriefed, and thanked for their participation.

## Data analysis

The extent of pressure induction was analysed using Bayesian One-Sided Paired Samples t-tests performed in JASP. Typing task performance was measured in WPM (the number of correct five-character strings typed per minute) and efficiency (correctly typed characters divided by total typed characters). Low pressure performance (WPM and efficiency) was determined as the average of the four low pressure trials (note that we retained one participant with three low pressure trials). To examine the effects of NDHCs on performance, we conducted a

Bayesian 2 Pressure (Mean Low Pressure vs. High Pressure) × 3 Contraction (NDHC, DHC, Sitting Quietly) ANOVA for both WPM and efficiency. We expected a main effect for Pressure and interaction between Pressure and Contraction on both ANOVAs.

## Results (Exp 3)

### Pressure manipulation

Bayesian One-sided Paired Samples t-tests revealed anecdotal evidence for an increase in SA and a decrease in S-C during the high-pressure condition (Table 11). There was moderate evidence for increased CA. A repeat analysis with left-handers and ambidextrous participants removed uncovered strong evidence for increased CA (BF10 = 19.63), and moderate evidence for increased SA (BF10 = 9.54) and decreased S-C (BF10 = 6.98).

### WPM

Descriptive statistics are in Table 12. The 2×3 Bayesian ANOVA on WPM revealed moderate evidence against a main effect of Pressure (Table 13), reflecting the minimal difference in WPM across pressure conditions (Table 12). There was anecdotal evidence against a main effect of Contraction condition and moderate evidence against the Pressure × Contraction interaction. In short, participants displayed similar WPM performance across all Pressure and Contraction conditions. No interpretative differences were found when removing left-handers/ambidextrous participants from the analysis.

### Efficiency analyses

The 2×3 ANOVA on Efficiency revealed anecdotal evidence for a main effect of Pressure (Table 13), with lower efficiency observed in the high-pressure condition (Table 12). We found moderate evidence against the Contraction main effect and anecdotal evidence against the Pressure × Contraction interaction. In short, participants' typing efficiency did not differ across Pressure and Contraction conditions.

## Discussion (Exp 3)

In Experiment 3, we explored the protective effects of NDHCs on typing performance under monitoring and competitive pressure. Our revised examination of NDHCs used a skill that was likely to be largely automated in our participants (Logan et al., 2011). We were also able to compare the effectiveness of NDHCs to DHCs in an ambidextrous task. Despite these adjustments, we did not observe performance differences across contraction conditions, and therefore, found no support for the beneficial effects of NDHCs.

## General discussion

In three experiments, we investigated the impact of NDHCs on the performance of common desktop activities under pressure. In Experiment 1, we examined if NDHCs assisted right-handed cursor positioning accuracy under time pressure. In Experiment 2, using the same

**Table 11. One-sided paired-samples t-test on MRF-L responses, M(SD).**

|  | **Low Pressure** | **High Pressure** | **BF$_{10}$** |
|---|---|---|---|
| Cognitive | 3.69 (2.71) | 4.51 (2.59) | 4.54 |
| Somatic | 3.87 (2.94) | 4.40 (2.73) | 1.17 |
| Confidence | 4.69 (2.64) | 5.29 (2.86) | 1.32 |

**Table 12. Performance descriptive statistics, M(SD).**

| Measure | Condition | Low Pressure | High Pressure |
|---------|-----------|--------------|---------------|
| WPM | NDHC | 78.55 (21.13) | 79.64 (19.40) |
| | DHC | 74.45 (20.39) | 75.81 (19.62) |
| | Sit Quietly | 77.17 (23.48) | 75.67 (23.24) |
| Efficiency | NDHC | .91 (.04) | .89 (.03) |
| | DHC | .91 (.04) | .91 (.04) |
| | Sit Quietly | .92 (.04) | .90 (.06) |

task, we examined if NDHCs facilitated cursor-positioning accuracy under evaluation and competition pressure. Finally, in Experiment 3, we studied if NDHCs assisted typing performance under evaluation and competition pressure. Across these studies, we found no evidence that NDHCs protect or facilitate performance under time or evaluative/competition pressure, despite an average increase in CA in each experiment. These results imply that NDHCs do not protect performance in simple, well-rehearsed, desktop motor skills performed under pressure.

Our findings differ from studies that imply that NDHCs protect performance in high pressure sport [9–11, 55]. Simultaneously, the findings echo recent investigations that did not find that NDHCs were beneficial to motor skill performance [56, 57]. A currently superficial understanding of how NDHCs achieve the specific behavioural outcome of improving motor performance under pressure is a challenge for understanding the mixed results.

Under the existing mechanistic logic, NDHCs should facilitate good performance in highly-rehearsed motor skills, because these skills function best with minimal verbal-analytic (conscious) input and NDHCs–by some means–reduce verbal-analytic interference in motor planning and execution [12]. Examining NDHCs in a high-pressure context should further help the elucidation of protective effects [57] because increased pressure may increase verbal-analytic motor processing and produce noisy "un-expert-like" brain states that have room for improvement. Given that we examined two kinds of highly rehearsed movements, and created anxiety in three studies, our results, along with the other recent null findings in this literature, implies that this mechanistic logic deserves further exploration.

## NDHCs in desktop motor skills vs sports skills

It may be easier to apply pressure to participants in sport-based studies than in everyday desktop tasks. For instance, while an elite tennis player might identify as an outstanding tennis player, a student who has incidentally learned to type well may not identify as a good typist or care much about that identity. Investment in a performance identity (e.g., being an athlete) is thought to encourage self-presentational concerns, for example, related to the broader perception of one's performance and ability [58]. Strong self-presentational concerns are considered

**Table 13. Bayesian 2 Pressure × 3 Contraction ANOVA for mean WPM and efficiency.**

| Measure | Effects | P(incl\|data) | BF_incl |
|---------|---------|---------------|---------|
| WPM | Pressure | .22 | .24 |
| | Contraction | .41 | .64 |
| | Pressure × Contraction | .03 | .36 |
| Efficiency | Pressure | .50 | 1.13 |
| | Contraction | .22 | .31 |
| | Pressure × Contraction | .05 | .44 |

central to experiencing pressure and performance anxiety [59] This may be especially true in sport, where the identity of elite athletes is often characterised by resilience under pressure [58]. Therefore, poor performance presents a threat to athletic identity, resulting in elevated anxiety [59].

Our participants may have had limited identity investment in cursor positioning or typing performance. Consequently, any self-presentational concerns or related anxiety might fall short of levels found in elite sport. However, we still observed an average increase in state anxiety after each manipulation, implying that pressure was created. To intensify the pressure on everyday skills in future research, highly skilful participants could be deliberately recruited to obtain a sample more sensitive to performance anxiety (e.g., typists who score above 90 words per minute on a pre-test). This could potentially elevate self-presentation concerns beyond motivational or monetary pressures. To provide a sharper analysis of the impact of pressure, future research could remove participants who did not experience elevated state anxiety under pressure, then proceed to analysis.

Notably, under current mechanistic understandings, it is equivocal that a pressure manipulation should even be necessary to observe beneficial effects of NDHCs. After all, NDHCs are supposed to promote an optimal brain state for motor skill execution, governed without verbal-analytic input. If NDHCs do in fact achieve this, performance should be facilitated in general, with or without pressure. To this end, we encourage future basic research–without pressure manipulations–examining if NDHCs can assist performance generally.

### Limitations

In this study, secondary measurements were not taken (e.g., EEG) to determine the neurological effect of NDHCs (e.g., reduced LH verbal-analytic processing). In our case, the online-only restriction prevented the use of EEG. It also prevented the use of the identical contraction apparatus (e.g., a stress ball). Instead, participants contracted a household item that virtually everyone has (socks). Contracting different items might have variable neurological impact, noting some evidence that contracting objects of higher internal pressure can promote greater asymmetrical hemispheric activation [60]. Additionally, the inability to control for environmental distractors (e.g. nearby conversations) could have impacted participants' performance and our findings. Similar future studies would ideally use controlled conditions, such as a darkened booth.

### Conclusion

This study explored the performance after-effects of NDHCs in computer-based motor-skills executed under pressure. Unlike previous work with expert sportspeople, these results did not support that NDHCs assist skilled movement under pressure. Our results challenge the utility of NDHCs as a treatment that can protect performance under pressure in highly rehearsed everyday motor tasks. Further research is needed to understand which movements, in which contexts, are likely to benefit from NDHCs.

### Author Contributions

**Conceptualization:** Yu Fan Eng, Daniel R. Little, Andy Yang, Anchalee Wensinger, Leo J. Roberts.

**Data curation:** Yu Fan Eng, Andy Yang.

**Formal analysis:** Yu Fan Eng.

**Investigation:** Yu Fan Eng, Andy Yang, Anchalee Wensinger.

**Methodology:** Yu Fan Eng, Daniel R. Little, Andy Yang, Anchalee Wensinger, Leo J. Roberts.

**Supervision:** Daniel R. Little, Leo J. Roberts.

**Writing – original draft:** Yu Fan Eng, Andy Yang, Leo J. Roberts.

**Writing – review & editing:** Daniel R. Little, Anchalee Wensinger, Leo J. Roberts.

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
