## [Decision Letter · Decision Letter 0]

14 Oct 2024

PONE-D-24-34528Non-dominant hand contractions do not facilitate performance under pressure in common desktop tasksPLOS ONE

Dear Dr. Roberts,

Thank you for submitting your manuscript to PLOS ONE. After careful consideration, we feel that it has merit but does not fully meet PLOS ONE’s publication criteria as it currently stands. Therefore, we invite you to submit a revised version of the manuscript that addresses the points raised during the review process. **Carefully review the referee's comments and revise your manuscript based on it and send it to the journal for re-review.** Please submit your revised manuscript by Nov 28 2024 11:59PM. If you will need more time than this to complete your revisions, please reply to this message or contact the journal office at plosone@plos.org. Please include the following items when submitting your revised manuscript:A rebuttal letter that responds to each point raised by the academic editor and reviewer(s). You should upload this letter as a separate file labeled 'Response to Reviewers'.A marked-up copy of your manuscript that highlights changes made to the original version. You should upload this as a separate file labeled 'Revised Manuscript with Track Changes'.An unmarked version of your revised paper without tracked changes. You should upload this as a separate file labeled 'Manuscript'.If applicable, we recommend that you deposit your laboratory protocols in protocols.io to enhance the reproducibility of your results. Protocols.io assigns your protocol its own identifier (DOI) so that it can be cited independently in the future. For instructions see: https://journals.plos.org/plosone/s/submission-guidelines#loc-laboratory-protocols. Additionally, PLOS ONE offers an option for publishing peer-reviewed Lab Protocol articles, which describe protocols hosted on protocols.io. Read more information on sharing protocols at https://plos.org/protocols?utm_medium=editorial-email&utm_source=authorletters&utm_campaign=protocols.

We look forward to receiving your revised manuscript.

Kind regards,

Rasool Abedanzadeh, Ph.D

Academic Editor

PLOS ONE

**Journal Requirements:**

**Additional Editor Comments:**

Dear authors,

Due to the passage of time since inviting several referees to evaluate the present manuscript, in order to avoid wasting more time, at this stage I decided to send you the result of the single referee. Carefully review the referee's comments and revise your manuscript based on it and send it to the journal for re-review.

Reviewers' comments:

Reviewer's Responses to Questions

**Comments to the Author**

1. Is the manuscript technically sound, and do the data support the conclusions?

Reviewer #1: Yes

2. Has the statistical analysis been performed appropriately and rigorously? 

Reviewer #1: Yes

3. Have the authors made all data underlying the findings in their manuscript fully available?

Reviewer #1: Yes

4. Is the manuscript presented in an intelligible fashion and written in standard English?

Reviewer #1: Yes

5. Review Comments to the Author

**Reviewer #1:** Thank you for the opportunity to review the manuscript with the title - Non-dominant hand contractions do not facilitate performance under pressure in common desktop tasks

Recommendations for improving the content of the manuscript:

Abstract

- we recommend adding relevant (numerical) results, not just the textual interpretation.

Introduction:

- The novel aspects of the present study should be detailed more concretely in relation to previous studies on the same topic.

Materials and Methods:

- The presentation of the two studies is detailed, but I think it could be compressed, the manuscript has 45 pages, which is much too long for an article.

Discussions:

- to add the practical implications based on the results of the study.

Conclusions - to compare the conclusions with 2-3 more relevant ideas based on the relevant results of the study and possibly to identify future research directions.

6. PLOS authors have the option to publish the peer review history of their article (what does this mean?). If published, this will include your full peer review and any attached files.

Reviewer #1: No

---

## [Author Response · Author response to Decision Letter 0]

18 Nov 2024

We thank the reviewer for considering our manuscript and their comments. 

we recommend adding relevant (numerical) results, not just the textual interpretation.

This has been addressed

Introduction:

- The novel aspects of the present study should be detailed more concretely in relation to previous studies on the same topic.

Our three studies test the after-affects of non-dominant hand contractions in common desktop motor tasks performed under pressure. To our knowledge, this has not been attempted before. Previous work has considered the context of elite music or sport.

We have edited the text starting line 151 to more concretely state this:

To date, researchers have considered the after-effects of NDHCs benefits in elite sport and music, with little consideration of the after-effects in non-elite domains. Accordingly, we tested the benefits of NDHCs on well-known (“everyday”) computer tasks conducted under time and/or performance pressure.

The presentation of the two studies is detailed, but I think it could be compressed, the manuscript has 45 pages, which is much too long for an article.

We shortened the manuscript throughout and have removed ~750 words. Please note that we have presented three studies rather than two.

- to add the practical implications based on the results of the study.

All studies found null results which limits the implications we can offer. It would be possible to suggest that NDHCs should not be used in desktop tasks performed under pressure, but this is not very practical. Had we found significant results we could have offered practical implications. Our results question a common finding in sport-based studies and suggest that the NDHC phenomenon requires more explanation and research (next point).

- to compare the conclusions with 2-3 more relevant ideas based on the relevant results of the study and possibly to identify future research directions.

We have suggested future research in a distributed fashion throughout the discussion on lines 658, 662, 669 and 681.

---

## [Editor Report · Decision Letter 1]

27 Nov 2024

PONE-D-24-34528R1Non-dominant hand contractions do not facilitate performance under pressure in common desktop tasksPLOS ONE

Dear Dr. Roberts,

Thank you for submitting your manuscript to PLOS ONE. After careful consideration, we feel that it has merit but does not fully meet PLOS ONE’s publication criteria as it currently stands. Therefore, we invite you to submit a revised version of the manuscript that addresses the points raised during the review process.

Please revise your manuscript according to the reviewer's comments and re-submit the revised manuscript.

We look forward to receiving your revised manuscript.

Kind regards,

Rasool Abedanzadeh, Ph.D

Academic Editor

PLOS ONE
---

## [Author Response · Author response to Decision Letter 1]

2 Dec 2024

We were asked to revise and resubmit, but no new review was attached to the correspondence. We have addressed the journal requirements noted (check references, and update figures with PACE software).

Kind regards, Leo

---

## [Editor Report · Decision Letter 2]

10 Dec 2024

Non-dominant hand contractions do not facilitate performance under pressure in common desktop tasks

PONE-D-24-34528R2

Dear Dr. Roberts,

We’re pleased to inform you that your manuscript has been judged scientifically suitable for publication and will be formally accepted for publication once it meets all outstanding technical requirements.

Kind regards,

Rasool Abedanzadeh, Ph.D

Academic Editor

PLOS ONE
---

## [Editor Report · Acceptance letter]

13 Dec 2024

PONE-D-24-34528R2 

PLOS ONE

Dear Dr. Roberts, 

I'm pleased to inform you that your manuscript has been deemed suitable for publication in PLOS ONE. Congratulations! Your manuscript is now being handed over to our production team.

Kind regards, 

on behalf of

Dr. Rasool Abedanzadeh 

Academic Editor

PLOS ONE